# Anti-Leukemia Activity of Polysaccharide from *Sargassum fusiforme* via the PI3K/AKT/BAD Pathway In Vivo and In Vitro

**DOI:** 10.3390/md21050289

**Published:** 2023-05-08

**Authors:** Haofei Du, Xudong Jin, Sizhou Jin, Donglei Zhang, Qiande Chen, Xuanan Jin, Caisheng Wang, Guoying Qian, Haomiao Ding

**Affiliations:** College of Biological and Environmental Sciences, Zhejiang Wanli University, Ningbo 315100, China; youzanzizwu@163.com (H.D.); jinxudong@zwu.edu.cn (X.J.); jsz2023010120@163.com (S.J.); zhangdonglei0621@163.com (D.Z.); chenqd0226@163.com (Q.C.); keling650555@163.com (X.J.); wangcs0528@163.com (C.W.)

**Keywords:** *Sargassum fusiforme*, polysaccharide, structural characteristic, leukemia, PI3K/AKT pathway, mitochondrial apoptosis

## Abstract

Studies have shown that *Sargassum fusiforme* and its extracts are effective herbal treatments for leukemia. We previously found that a polysaccharide from *Sargassum fusiforme*, SFP 2205, stimulated apoptosis in human erythroleukemia (HEL) cells. However, the structural characterization and antitumoral mechanisms of SFP 2205 remain uncertain. Here, we studied the structural characteristics and anticancer mechanisms of SFP 2205 in HEL cells and a xenograft mouse model. The results demonstrated that SFP 2205, with a molecular weight of 41.85 kDa, consists of mannose, rhamnose, galactose, xylose, glucose, and fucose with monosaccharides composition of 14.2%, 9.4%, 11.8%, 13.7%, 11.0%, and 38.3%, respectively. On animal assays, SFP 2205 significantly inhibited growth of HEL tumor xenografts with no discernible toxicity to normal tissues. Western blotting showed that SFP 2205 therapy improved Bad, Caspase-9, and Caspase-3 protein expression, and ultimately induced HEL tumor apoptosis, indicating mitochondrial pathway involvement. Furthermore, SFP 2205 blocked the PI3K/AKT signaling pathway and 740 Y-P, an activator of the PI3K/AKT pathway, rescued the effects of SFP 2205 on HEL cell proliferation and apoptosis. Overall, SFP 2205 may be a potential functional food additive or adjuvant for preventing or treating leukemia.

## 1. Introduction

Cancer is the top cause of mortality worldwide and a significant impediment to extending life expectancy [1,2,3]. Since 1975, the total cancer incidence rate has increased by 0.8% annually in children and adolescents; however, trends differ by cancer type. Leukemia accounts for 28% of childhood cancer diagnoses, followed by brain and other nervous system tumors (26%), of which, nearly one-third are benign or borderline malignant [4]. Leukemia originates from the hematopoietic system and is a malignant clonal disease with high heterogeneity. Its main feature is clonal proliferation of primitive white blood cells in the bone marrow, peripheral blood, and other organs and tissues, which adversely affects the bone marrow and lymphatic system in humans and leads to high mortality [5]. Chimeric antigen receptor T cells and molecular-targeted treatments are used to treat leukemia in addition to traditional chemotherapy and hematopoietic stem cell transplantation [6]. However, the high costs associated with creation of these targeted medications have led to the need for novel, low-cost alternatives, such as pharmacologically active compounds derived from natural sources [7].

Research has shown that natural active substances derived from microorganisms, marine organisms, plants, and fungi exhibit diverse pharmacological and biological properties. Voacamine, a bisindole alkaloid isolated from *Voacanga africana* Stapf, stimulates apoptosis in breast cancer cells by activating the mitochondrial-associated apoptosis signaling pathway and suppressing the PI3K/AKT/mTOR signaling pathway. Furthermore, it significantly reduces tumor size without displaying appreciable toxicity [8]. In HepG2 cells, peptides from silkworm pupae (*Bombyx mori*) exhibited great antioxidant activity (i.e., reactive oxygen species reduction, superoxide dismutase expression, and glutathione generation) [9]. *Morchella esculenta* polysaccharide had a protective effect on dextran-sodium-sulfate-induced liver injury, and its mechanism may be related to reducing oxidative stress, inhibiting inflammatory response and increasing the activity of liver antioxidant enzymes [10]. Agents such as *Sargassum fusiforme* polysaccharides can effectively suppress cancer cell proliferation and stimulate apoptosis [11]. The biological activity of polysaccharides is directly tied to their structure. We previously extracted a polysaccharide from *S. fusiforme* with a mean molecular weight of 299 kDa [12]. This molecular weight is too large to be used by organisms; therefore, we separated out a polysaccharide with a smaller molecular weight.

*S. fusiforme* is a prevalent brown alga along rocky shores in Korea, Japan, and China (particularly in the Zhejiang and Fujian provinces). *S. fusiforme* has been used as a traditional Chinese herb for treating thyroid problems for thousands of years [13]. *S. fusiforme* polysaccharide, a sulfated polysaccharide, is a semisynthetic chemical product containing natural polysaccharide derivatives with sulfate groups. It contains many biological properties, including antiviral, anticoagulative, antitumoral, and antioxidant properties, and it may boost the immunological system [14,15,16,17,18]. Studies have established that the active sites of cysteine-containing aspartate proteolytic enzymes (caspases) contain cysteine residues that can cleave peptide bonds on aspartate residues in target proteins [19]. These sites are responsible for selective cleavage of certain proteins, resulting in apoptosis [20,21]. Caspases have become targets for designing new antitumoral drugs and treatments, and several studies have reported that natural components exhibit mechanisms with antitumoral activity to activate the caspase signaling pathway. Our prior study showed that *S. fusiforme* polysaccharide significantly enhanced Bax, Bad, and Caspase-3 expression in human erythroleukemia (HEL) cells via the caspase signaling pathway [11]. However, whether *S. fusiforme* polysaccharide stimulates HEL cell apoptosis via the caspase signaling pathway in vivo and plays an anti-erythroleukemia role remains unknown.

Here, we extracted a new polysaccharide fraction from *S. fusiforme* and determined its physicochemical features, such as its molecular weight, monosaccharide composition, and sulfation degree. We used HEL cells and a xenograft model to evaluate the anticancer and antiproliferative effects of this *S. fusiforme* polysaccharide. Our results may provide a scientific basis for using *S. fusiforme* polysaccharide as a health-maintenance agent with anticancer properties.

## 2. Results

### 2.1. Molecular Weight of SFP 2205

Figure 1 shows the purity and molecular weight results for SFP 2205 as determined by HPGPC. Compared with the control group (Figure 1A), SPF 2205 displayed a single symmetrical peak at 23.93 min (Figure 1B), indicating that SFP 2205 is relatively uniform with high purity. The molecular weight of SFP 2205 was determined via dextran molecular weight standards (500 kDa, 100 kDa, 70 kDa, 40 kDa, and 10 kDa). From the standard curve equation, y = −0.196× + 9.312 (R^2^ = 0.9954), the molecular weight of SFP 2205 was determined to be 41.85 kDa.

### 2.2. UV and Infrared Spectroscopic Analysis of SFP 2205

The prepared SFP 2205 polysaccharide aqueous solution was scanned on a UV-visible spectrophotometer from a wavelength of 200–500 nm. Distilled H_2_O was used as a negative control. The SFP 2205 aqueous solution absorbed light at 215 nm, indicating that it contained polysaccharide (Figure 2A). Additionally, the absence of absorption at 260 and 280 nm in the UV spectrum (Figure 2A) suggested that SFP 2205 lacked proteins or nucleic acids.

The infrared spectrum scanning results for SFP 2205 (Figure 2B) revealed the following. The detected sample had a characteristic vibrational absorption peak of carbohydrates at 3000–2800 cm^−1^, a strong O-H stretching vibration absorption peak, and an intramolecular or intermolecular hydrogen bond. C-H had a stretching vibration absorption peak at 2933 cm^−1^. A possible C=O stretching vibration was noted in the amide group at 1633 cm^−1^, and a characteristic absorption of the α-pyran ring occurred at 754.6 cm^−1^. The vibrational peaks of C-O-C and C-OH occurred at 1090–1260 cm^−1^. SFP 2205 peaked at 1256.99 cm^−1^, showing the symmetric stretching vibration of S=O of the sulfate radical. SFP 2205 has a characteristic carbohydrate structure.

### 2.3. SFP 2205 Monosaccharide Composition and Sulfate Group Content

HPLC revealed SFP 2205’s monosaccharide composition. SFP 2205 contained the following heteropolysaccharides and corresponding molar percentages: mannose (14.2%), rhamnose (9.4%), galactose (11.8%), xylose (13.7%), glucose (11.0%), and fucose (38.3%) (Figure 3A,B). Thus, SFPS 2205 is a highly pure polysaccharide extract containing fucose.

We used the barium sulfate turbidity method, using K_2_SO_4_ as the standard product, the absorbance difference (A_1_ − A_2_) as the ordinate, and the sulfate group content of the standard product as the abscissa to develop a standard curve (Figure 3C). The regression equation was y = 2.2071× − 0.0192, R^2^ = 0.9932. Using the (A_1_ − A_2_) sample absorbance, the above equation was substituted to obtain the mass of the sulfate group in the polysaccharide. Finally, the sulfate content in SFP 2205 was calculated as 19.64%.

### 2.4. NMR Analysis of SFP 2205

NMR spectroscopy provides precise structural information on carbohydrates, such as monosaccharide compositions, linkage patterns, and sugar unit sequences in polysaccharides. A ^1^H chemical shift was evident (Figure 4A). The ^1^H NMR (600 MHz, D_2_O) spectrum of SFP 2205 was δ 5.50–4.77 (m, 29H), 5.28–5.21 (m, 4H), 5.28–4.77 (m, 22H), 5.09 (d, *J* = 87.8 Hz, 18H), 4.71 (s, 8H), 4.65 (s, 136H), 4.51–4.27 (m, 11H), 4.51–4.13 (m, 19H), 4.51–3.83 (m, 39H), 4.55–2.86 (m, 74H), 3.83–2.86 (m, 34H), 3.24 (d, *J* = 49.1 Hz, 3H), 3.24 (d, *J* = 49.1 Hz, 3H), 3.24 (d, *J* = 49.1 Hz, 3H), 3.24 (d, *J* = 49.1 Hz, 3H), 4.05–0.76 (m, 94H), 2.46–1.47 (m, 5H), 2.46–1.55 (m, 5H), 2.46–1.54 (m, 5H), 2.46–1.47 (m, 5H), 2.79–0.73 (m, 43H), and 1.47–0.81 (m, 27H). Hence, the anomeric hydrogens at 5.09, 4.71, 4.65, and 4.61 ppm indicated that SFP 2205 contained α- and β-configurations. The NOESY spectrum of NMR revealed a methyl double peak with a 6-position deoxy sugar at 1.1–1.3 ppm, 4 methyl peaks with acetyl amino at 2.0–2.2 ppm, and a very small region at 3.0–4.2 ppm, where the results overlapped too heavily to yield structural information (Figure 4B).

### 2.5. Effect of SFP 2205 Treatment on Tumor Volume and Weight

Our prior studies demonstrated that SFP 2205 significantly prevented HEL cell development in vitro [11]. Here, we aimed to determine whether SFP 2205 inhibits tumor cell growth in HEL tumor-bearing mice in vivo. Tumor growth was decreased but not completely eliminated in the high-concentration SFP 2205 group, demonstrating that SFP 2205 alone could not completely prevent tumor growth (Figure 5A). The polysaccharide concentration in SFP 2205 was primarily responsible for its antitumor activity. To further compare treatment efficacy, we determined tumor growth inhibition (TGI), relative tumor volume (RTV), and relative tumor proliferation rates for each group. The relative tumor proliferation rate is shown as T/C (%). Mice in the low-concentration SFP 2205 group were exposed to a TGI (%) of 20.29 ± 14.48% and a T/C (%) of 93.76 ± 12% (Figure 5B,C). Mice in the middle-concentration group showed a TGI (%) of 53.39 ± 15.31% and a T/C (%) of 45.32 ± 10.23%. Mice in the high-concentration group received the greatest antitumor effectiveness, with a TGI (%) of 79.29 ± 6.84% and a T/C (%) of 29.56 ± 10.84%. In the PBS and low-concentration SFP 2205 groups, tumor growth was not significantly suppressed; however, the PBS group exhibited a higher tumor growth rate than did the low-concentration SFP 2205 group. Mice in the middle- and high-concentration SFP 2205 groups exhibited anticancer effects (Figure 5D–F). The body weights of the mice did not differ between the control and treatment groups, indicating that the SFP 2205 doses administered caused no discernible toxicity in the mice (Figure 5G). All mice in the high-concentration SFP 2205 group survived for >40 days, which was significantly longer than the mice in the other groups (Figure 5H). Thus, the high-concentration SFP 2205 was the most efficient at suppressing tumor growth and extending survival.

### 2.6. Effects of SFP 2205 on Main Organ and Tumor Morphology in HEL Tumor-Bearing Mice

H&E staining was used to examine major organ and tumor morphologies (Figure 6). H&E stains nucleic acids dark blue-purple and cytoplasm pink, allowing tumor tissue structure and morphological characteristics to be evaluated. In the control group, tumor cells were structurally intact and well-arranged with unambiguous mitosis, demonstrating a proliferative environment, whereas the SFP 2205-treated groups exhibited partial nucleic pyknosis, a larger cytoplasmic area, and fatty degeneration (Figure 6A).

To investigate whether SFP 2205 treatment would cause toxicity to organs other than the tumor tissue in HEL-tumor-bearing mice, key organs (heart, liver, spleen, lungs, and kidneys) were stained with H&E (Figure 6B). SFP 2205 treatment had minimal pathogenic effects, such as intracellular space and vacuolization, in the liver. Overall, organ samples treated with SFP 2205 exhibited no clear morphological or structural changes, pathological damage, aberrant lines, or visible lesions, thereby indicating no systemic organ damage.

### 2.7. Effects of SFP 2205 on Expression of Apoptotic Pathway-Related Genes and Proteins

qRT-PCR was used to evaluate apoptotic gene expression of *Bad*, *Bcl-xL*, *Caspase-9*, and *Caspase-3* in the mitochondria-dependent pathway to further examine potential mechanisms of SFP 2205-induced apoptosis in HEL tumors. SFP 2205 treatment significantly upregulated expression of the proapoptotic gene, *Bad*, and significantly downregulated expression of the antiapoptotic gene, *Bcl-xL* (*p* < 0.05), compared with those of the control group (Figure 7A,B). Similarly, *Caspase-9* gene expression increased concentration-dependently relative to that of the control group, leading to a concentration-dependent increase in downstream *Caspase-3* gene expression (*p* < 0.05; Figure 7C,D). Overall, gene expression linked with the mitochondrial apoptotic pathway demonstrated that SFP 2205 induced HEL tumor apoptosis via the mitochondrial signaling pathway.

To further study the probable mechanisms of SFP 2205-induced apoptosis in HEL tumors, apoptotic protein expression of Bad, Bcl-xL, Caspase-9, and Caspase-3 in the mitochondria-dependent pathway was assessed through Western blotting (Figure 7E). SFP 2205 treatment downregulated antiapoptotic protein expression (Bcl-xL) and significantly upregulated proapoptotic protein expression (Bad; *p* < 0.05) compared with that of the control group (Figure 7F,G). Similarly, Caspase-9 protein expression was elevated concentration-dependently relative to that of the control group (*p* < 0.05; Figure 7H). Additionally, Caspase-3 expression increased dramatically and dose-dependently compared with that of the control group (*p* < 0.05; Figure 7I), resulting in apoptosis. Overall, protein expression linked with the mitochondrial apoptotic pathway demonstrated that SFP 2205 induced HEL apoptosis via mitochondrial signaling.

### 2.8. SFP 2205 Regulated Activity of the PI3K/AKT-Mediated Signaling Pathway in HEL Cells

Because the PI3K/AKT pathway is abnormally active in numerous malignancies and is closely associated with tumor proliferation progression, we hypothesized that the PI3K/AKT signaling pathway may mediate the effects of SFP 2205 on HEL cell proliferation. 740Y-P, an activator of the PI3K/AKT pathway, was used in a series of rescue studies to validate the role of the PI3K/AKT pathway. A CCK-8 assay was used to observe viability in HEL cells exposed to SFP 2205 or 740 Y-P. In Figure 8A, the SFP 2205 reduced the viability of HEL cells after 24 h treatments in a dose-dependent manner. The IC_50_ (the concentration of SFP 2205 that reduced cell viability by 50%) was 55.38 ± 4.31 μg/mL, which will be used for future experiments. After pretreatment with 740 Y-P (30 μg/mL) for 1.5 h, cell viability was 110.43% ± 1.20%, indicating that 740 Y-P pretreatment substantially increased HEL cell viability (*p* < 0.05; Figure 8B). SFP 2205 significantly suppressed HEL cell viability, whereas the PI3K agonist, 740 Y-P, partially rescued this effect, indicating that 740 Y-P decreased the antitumor effect of SFP 2205 in HEL cells.

To explore SFP 2205’s possible role in cell proliferation, its effect on the cell cycle and apoptosis in HEL cells was assessed using flow cytometry. Cell apoptosis assays were evaluated by annexin-V/PI staining and showed that SFP 2205 enhanced the apoptosis rate of HEL cells compared with that of the control group (Figure 8C,E). After SFP 2205 treatment, cell percentages were increased in the G0/G1 phase and reduced in the S phase (Figure 8D,F). These findings suggest that SFP 2205 suppressed cell proliferation by blocking cell cycle progression and stimulating HEL apoptosis. 

### 2.9. PI3K/AKT/Bad/Bcl-xL Axis Contributed to SFP 2205-Induced Apoptosis

To further investigate the mechanism underlying SFP 2205-induced apoptosis, we analyzed SFP 2205’s effect on expression of classic apoptosis-related pathways, including Bad/Bcl-xL and PI3K/AKT signaling. PI3K gene expression levels significantly increased in the 740 Y-P group compared with the control group, and pretreatment with 740 Y-P reversed gene-level changes in *PI3K* in HEL cells incubated with SFP 2205 (Figure 9A). HEL cells treated with SFP 2205 exhibited no significant changes in *AKT* expression compared with that of the control group (Figure 9B). Furthermore, 740 Y-P pretreatment downregulated gene expression levels of *Bad*, *Caspase-9*, and *Caspase-3* and upregulated gene expression of *Bcl-xL* in HEL cells incubated with SFP 2205 (Figure 9C–F). Next, protein expression of the PI3K/AKT/Bad/Bcl-xL pathway was tested (Figure 9G,H). PI3K protein expression levels significantly increased in the 740 Y-P group compared with the control group, and pretreatment with 740 Y-P reversed protein-level alterations of PI3K in HEL cells incubated with SFP 2205 (Figure 9I). HEL cells treated with SFP 2205 exhibited no significant changes in AKT protein expression compared with that of the control group (Figure 9J). *p*-AKT protein expression levels were significantly decreased in the control group. Pretreatment with 740 Y-P (PI3K agonist) inhibited downregulation of SFP 2205-induced *p*-AKT protein expression (Figure 9K). Pretreatment with 740 Y-P reversed protein-level changes in Bad, Bcl-xL, Caspase-9, and Caspase-3 in HEL cells incubated with SFP 2205 (Figure 9L–O). These data suggest that SFP 2205 promoted apoptosis in HEL cells by suppressing PI3K/AKT pathway activity.

## 3. Discussion

Studies have shown that some natural products may enhance tumor occurrence and development via processes involving cytotoxicity, cell cycle arrest, mitochondrial disruption, and the nitric-oxide-dependent pathway [22]. Previous studies showed that polysaccharides from brown algae possess anti-inflammatory, antiphotoaging, antioxidant, antidiabetic, and antitumor activity [15,23,24,25]. Brown algal polysaccharides exhibit antitumor action and can greatly reduce cancer cell proliferation, suggesting their potential as a safe, natural anticancer therapy. Polysaccharides extracted from *Laminaria japonica* have been found to inhibit leukemia progression [26]. Comparative apoptosis experiments showed that the type II polysaccharide extracted from *Fucus vesiculosus* induced apoptosis via Caspase-8 and Caspase-9 activation in MCF-7 and HeLa cells in a manner comparable to that of low-molecular-weight type I polysaccharide derivatives [27]. Species-related structural variation, growth circumstances, and extraction processes make it difficult to determine the anticancer activity of polysaccharides from brown algae [28]. Here, we extracted and characterized polysaccharides from brown algae (*Sargassum fusiforme*). Monosaccharide composition analysis showed that SFP 2205 consists of mannose, rhamnose, galactose, xylose, glucose, and fucose. The fucose content was comparable to that described in the literature, but the mannose, galactose, xylose, and glucose contents differed significantly from those of Cong et al. [29], and rhamnose could not be determined. An extremely intense broadband at 1250 cm^−1^ was ascribed to asymmetric O=S=O stretching vibration of sulfate esters, with contributions from COH, CC, and CO vibrations, whereas the band at 842 cm^−1^ was attributed to sulfate groups at axial C4 positions, indicating that sulfation occurred at the C4 positions. Band absence at 820 cm^−1^ was ascribed to C-O-S bending vibration of sulfate substituents at equatorial C2 or C3 positions, suggesting that little or no sulfation occurred at C2 or C3 [30]. Hence, most SFP 2205 sulfation occurred at the C4 position. SFP 2205 contained 26.56% ester sulfate. Similar to other brown seaweed polysaccharides, SFP 2205 exhibited overlapping resonances in its ^1^H and NOESY spectra, mostly due to its complex glycosyl composition and various sulfation substitutions. 

Balb/c nude mice with subcutaneous HEL tumors were used as a treatment model. We assessed its medicinal impacts in Balb/c nude mice to verify whether SFP 2205 could inhibit HEL tumor growth in vivo. After 12 days of therapy, the RTV indicated that SFP 2205 significantly inhibited HEL xenograft tumors compared with those of the non-tumor group. Polysaccharides from *Sargassum fusiforme* are a family of sulfated polysaccharides. One study showed that 50, 100, and 200 mg/kg of crude polysaccharides from *Sargassum fusiforme* inhibited nasopharyngeal cancer growth by 18%, 28%, and 43%, respectively [8]. Moreover, 20 and 40 mg/kg *Sargassum fusiforme* polysaccharide reduced SPC-A-1 cell proliferation [3]. Every polysaccharide was a crude polysaccharide. The polysaccharides were then isolated and purified to identify their active ingredients. Here, we isolated and described a newly discovered sulfated polysaccharide, SFP 2205, and determined its anti-leukemia activities in vivo. HEL xenograft tumor model results confirmed that SFP 2205 had high anticancer activity. 

The intrinsic apoptotic route is a conventional apoptotic pathway primarily controlled by the Bcl-2 family, which includes the antiapoptosis protein, Bcl-xL, and the proapoptosis protein, Bax [31,32,33]. Research has shown that when cells undergo outside stimulation, Bad is upregulated and Bcl-xL is downregulated, which impedes the mitochondrial membrane, thus activating the caspase family and triggering an apoptotic response [34,35,36]. Studies have demonstrated that polysaccharides derived from brown algae stimulate cancer cell apoptosis in various ways. Wang et al. found that polysaccharide from a Celluclast-assisted extract of an edible brown seaweed (*Sargassum fulvellum*) considerably increased mRNA levels of apoptosis-related genes and significantly elevated Bax and cleaved Caspase-3 protein expression in vitro [37]. Fernando et al. investigated the mechanism of inhibition of HL-60 cells by isolating polysaccharide from *Sargassum polycystum*. *Sargassum polycystum* polysaccharide upregulated Bax, Caspase, and p53 proteins and stimulated cell apoptosis [38]. We discovered the apoptosis mechanism of *Sargassum fusiformis* polysaccharide via Western blot and found that it induced tumor tissue apoptosis by upregulating Bad and downregulating Bcl-xL, thus activating Caspase-3 and ultimately leading to apoptosis. 

The PI3K/AKT signaling pathway is important for controlling protein synthesis and regulating cell proliferation, differentiation, and migration [39,40,41]. This pathway is overactivated in numerous cancers, including leukemia [42,43,44], and inhibits apoptosis induced by several stimuli, enhances proliferation, and contributes to metastasis and tumor blood vessel formation [45,46,47]. By inhibiting survival pathways and inducing apoptosis in cancer cells, PI3K/AKT pathway targeting may be effective against malignancy [48,49,50]. Ran et al. [51] reported that pollen polysaccharides from Chinese wolfberry promote prostate cancer DU145 cell apoptosis, and the antitumor mechanism on DU145 cells may regulate the PI3K/AKT signaling pathway, which ultimately increases apoptosis. Corn silk crude polysaccharide inhibits the EGFR/PI3K/AKT/CREB signaling pathway to exert its antipancreatic cancer activity [52]. Accordingly, we hypothesized that SFP 2205 could trigger apoptosis in HEL cells via the PI3K/AKT pathway. Our results demonstrated that SFP 2205 treatment significantly lowered *p*-AKT expression levels in HEL cells. As a result of SFP 2205 inhibition of AKT phosphorylation, Bad was translocated into the mitochondria, forming a proapoptotic complex with Bcl-xL, which enhanced cytochrome release and induced apoptosis. SFP 2205 upregulated Bad expression, resulting in cell cycle arrest in the G0/G1 phase in erythroleukemia cells. Thus, SFP 2205 may stimulate apoptosis by suppressing the PI3K/AKT/Bad/Bcl-xL/Caspase-9/Caspase-3 signaling pathway in erythroleukemia cells. Piceatannol availably suppressed activity and proliferation and induced apoptosis of osteosarcoma cells. Conversely, 740Y-P reversed the effects of piceatannol on osteosarcoma cells, which promoted cell activity and proliferation and restrained apoptosis [53]. We further examined PI3K/AKT pathway involvement in SFP 2205 using 740Y-P, an activator of the PI3K/AKT signaling pathway. We discovered that 740Y-P partially reversed the effects of SFP 2205 on HEL involvement and molecular expression. Thus, we hypothesize that SFP 2205 inhibits cell proliferation and enhances cell apoptosis via the PI3K/AKT signaling pathway in HEL cells cultured in vitro.

## 4. Materials and Methods

### 4.1. Preparation and Purification of S. fusiforme Polysaccharide

*S. fusiforme* was obtained from Dongtou, Zhejiang Province, China. At 87 °C, 250 g of algal powder was processed with water (1:35, *w*/*v*) and removed for 13 min at 900 W. The precipitate was obtained via centrifugation, and protein was extracted as described previously [54]. The deproteinated supernatant was then lyophilized to produce crude polysaccharides. Using DEAE-FF cellulose (1.6 cm × 20 cm, Zhongsen, Suzhou, China) and Sephacry^TM^ S-200 HR columns (1.6 cm × 20 cm, GE Healthcare, Uppsala, Sweden), crude extracts were purified to yield the *S. fusiforme* polysaccharide SFP 2205, which was lyophilized for additional studies.

### 4.2. SFP 2205 Molecular Weight

The SFP 2205 molecular weight was assessed and determined using high-performance gel permeation chromatography (HPGPC). Sample solution was applied to a Waters high-performance liquid chromatography (HPLC) system equipped with a TSK-GEL G4000 PWXL column (7.8 mm × 300 mm, TOSOH, Tokyo, Japan), eluted with 0.1 mol/L Na_2_SO_4_ solution at a flow rate of 0.4 mL/min, and determined using an Agilent 1620 Refractive Index Detector (Waldbronn, Germany). Columns were calibrated using dextran T-series standards (T-500, T-100, T-70, T-40, and T-10) of known molecular weight (Shanghai, China). The SFP 2205 molecular weight was calculated with reference to the above calibration curve.

### 4.3. Ultraviolet (UV) and Infrared (IR) Analysis of SFP 2205

SFP 2205 (10 mg) was hydrolyzed with 5 mL distilled water, and UV absorption spectra were documented on a UV-2600 spectrometer (Shimadzu, Kiya-cho, Japan) in the range of 200–500 cm^−1^. Using a Vertex 70 spectrometer, the Fourier transform (FT)-IR spectrum of SFP 2205 was determined to explore its functional groups (Bruker Optics, Ettlingen, Germany). Each sample (2 mg) was combined with 200 mg of KBr powder and compressed to form a pellet. FT-IR spectra were documented between 4000 and 400 cm^−1^ in wavelength.

### 4.4. Preliminary Characterization of SFP 2205

Ester sulfate from SFP 2205 was measured as per Yu et al. [55]. The SFP 2205 monosaccharide composition was investigated using HPLC. SFP 2205 was initially hydrolyzed for 4 h at 100 °C with 4 M trifluoracetic acid. After hydrolysis, monosaccharides were identified using an HPLC system with a Hi-Plex Ca column (7.7 mm × 50 mm, Agilent, Santa Clara, CA, USA) and a refractive index detector. Reference samples of mannose, rhamnose, galactose, xylose, glucose, and fucose (Sigma, St. Louis, MO, USA) were used as monosaccharide standards.

### 4.5. Nuclear Magnetic Resonance (NMR) Spectroscopy

NMR spectroscopy was performed as previously described [56]. By lyophilization, 140 mg SFP 2205 was co-evaporated twice with D_2_O, then dissolved in 0.5 mL D_2_O. ^1^H NMR were recorded with a 600 MHz spectrometer (Agilent Technologies, Santa Clara, CA, USA). Two-dimensional nuclear Overhauser effect spectroscopy (NOESY) was obtained at 600 MHz and 25 °C with sufficient acquisition time.

### 4.6. Animal Handling and Establishing the HEL-Bearing Mouse Model

All Balb/c mice were purchased from Shanghai Model Organisms Technology (Shanghai, China) and housed in the animal center of Ningbo University, China. Mice were housed 4 per cage at 21 ± 2 °C, 50 ± 10% relative humidity, and a 12 h light/12 h dark cycle and received food and water ad libitum. The National Institutes of Health Guide for Care and Laboratory Animal Use and Protection of Ningbo University Health Science Center governed all experimental methods (protocol code NBU-2022-11749, approval date 16 August 2022).

After 7 days of adaptation, 2 × 10^6^ HEL cells resuspended in 100 μL Matrigel basement membrane matrix (Solaibo, Beijing, China) were subcutaneously injected into the right flank of each mouse to generate tumors. Balb/c mice were randomly separated into the physiological (phosphate-buffered) saline (PBS) group and three SFP 2205 groups (*n* = 12 per group). When the primary tumors reached 30 mm^3^, mice were treated via tail vein injection of 100 μL physiological saline, 100 μL SFP 2205 at 20 mg/kg (low-concentration group), 100 μL SFP 2205 at 40 mg/kg (middle-concentration group), or 100 μL SFP 2205 at 80 mg/kg (high-concentration group), every 2 days for 12 days. The largest transverse diameter (width) and greatest longitudinal diameter (length) of the tumors were measured daily with a vernier caliper to investigate the tumor progression. Tumor volume was estimated as tumor volume = (tumor length) × (tumor width)^2^/2. The tumor-bearing mice were weighed every 2 days.

### 4.7. Paraffin Sectioning and Hematoxylin and Eosin (H&E) Staining

The hearts, livers, spleens, lungs, kidneys, and tumor tissues from the tumor-transplanted mice were fixed overnight in 4% paraformaldehyde, then transferred to 70% ethanol. Tissues were then dehydrated and transparentized in increasing concentrations of ethanol and xylene, fixed in paraffin, sectioned to a thickness of 5 μm, dewaxed with xylene, and stained with H&E (Beyotime Biotechnology, Shanghai, China). The stained tissue sections were viewed under a light microscope (Nikon, Tokyo, Japan).

### 4.8. Cell Lines and Cultures

HEL cells were obtained from the Shanghai Institute of Cell Biology of the Chinese Academy of Sciences(Shanghai, China). HEL cells were stored in RPMI-1640 media with antibiotics (100 IU/mL penicillin and 100 μg/mL streptomycin) and 10% fetal bovine serum at 37 °C in 5% CO_2_ humidified air. HEL cells were incubated at 5 × 10^3^ cells per well in 96-well plates. When cultivated cells reached 80% confluence, the medium was changed to RPMI-1640 with various concentrations of SFP 2205 or 30 μg/mL 740 Y-P and cultivated for another 24 h. Before adding fresh media to the cell cultures, the medium was filtered through 0.45 μm filters.

### 4.9. Analysis of Apoptosis

Apoptosis was quantified using a kit (BD, Tokyo, Japan) per the manufacturer’s protocol as previously described. HEL cells were stained with fluorescein isothiocyanate/propidium iodide (FITC/PI). Cell apoptosis rates were evaluated using a flow cytometer (BD FACSVerse, Franklin Lakes, NJ, USA).

### 4.10. Cell Cycle Analysis via Flow Cytometry

Cell cycles were analyzed via PI staining. After treatment with SFP 2205 or 740 Y-P, HEL cells were collected, rinsed with sterilized cold PBS, and fixed with cold 75% ethanol overnight. Fixed cells were incubated with PI in the dark for 10–15 min, then flow cytometry analysis was performed.

### 4.11. Quantitative Real-Time-PCR (qRT-PCR) for Gene Expression

Total RNA from HEL tumor cells after SFP 2205 or 740 Y-P treatment for 24 h were used for qRT-PCR. Less than 1 μg of total RNA was reverse transcribed and measured using a SYBR Green RT-PCR Master Mix Kit (Tokyo, Japan). β-actin was used as an internal standard. Table 1 lists the primers used. Relative expression of each gene was determined and normalized using the 2^−ΔΔCt^ method relative to β-actin. Each experiment was conducted three times.

### 4.12. Western Blot Assays

Tumor tissues and HEL cells from each group were rinsed with PBS and lysed using cell lysis buffer containing phosphatase inhibitor and protease inhibitor (Solaibo, Beijing, China). After centrifugation and collection of the supernatants, protein quantities were measured and normalized. Protein samples were blended with SDS-PAGE sample loading buffer (Solaibo, Beijing, China) and heated at 95 °C for 5 min.

Protein samples were separated on a 10% SDS-PAGE gel, then transported to nitrocellulose membranes (Millipore Co., Billerica, MA, USA) and blocked for 2 h at room temperature with 3% non-fat dry milk in TBST (Tris Buffered Saline with Tween 20). Membranes were rinsed with TBST and incubated at 4 °C overnight after adding primary antibodies. The primary antibodies used were GAPDH (1:2000, Cell Signaling Technology, Beverly, MA, USA), PI3K (1:1000, Cell Signaling Technology), AKT (1:1000, Abcam, Waltham, MA, USA), *p*-AKT (1:1000, Cell Signaling Technology), Bad (1:1000, Cell Signaling Technology), Bcl-xL (1:1000, Cell Signaling Technology), Caspase-9 (1:1000, Cell Signaling Technology), and Caspase-3 (1:1000, Cell Signaling Technology). Membranes hybridized to primary antibodies were rinsed with TBST, then incubated for 2 h at room temperature after adding a horseradish peroxidase-conjugated goat anti-rabbit IgG antibody (1:4000, Abclonal Technology, Wuhan, China). After rinsing the membranes with TBST, the chemiluminescent substrate Omni-ECLTM Kit (EpiZyme Biomedical Technology, Shanghai, China) was applied, and images were recorded using an XRS camera fitted with a Tanon imaging system. The Western blots were quantified and evaluated using ImageJ (version 1.8.0, Bethesda, MD, USA).

### 4.13. Statistical Analysis

All data are presented as means ± standard deviation. One-way analysis of variance was used to evaluate statistically significant differences between experimental groups, and differences were considered statistically significant if *p* < 0.05. All computations were performed using SPSS 16.0 statistical software.

## 5. Conclusions

SFP 2205 reduced HEL cell growth in HEL tumor-bearing mice while showing no clear toxicity to the mice. Additionally, SFP 2205 activated HEL cell apoptosis in vitro and in vivo. The anti-leukemia mechanism of apoptosis stimulated by SFP 2205 may involve intrinsic mitochondrial caspase stimulation and PI3K/AKT signaling pathway suppression. Moreover, anticancer action on HEL cells triggered G1 cell cycle arrest by modulating cell-cycle-related proteins. SFP 2205 should be researched further for use as a potential safe food additive for preventing leukemia.

## Figures and Tables

**Figure 1 marinedrugs-21-00289-f001:**
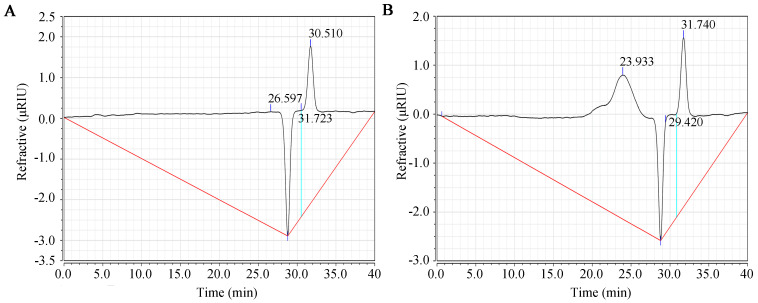
Molecular weight distribution of *Sargassum fusiforme* polysaccharide (SFP 2205). (**A**) High-performance gel permeation chromatography (HPGPC) chromatogram of the control group. (**B**) HPGPC chromatogram of SFP 2205.

**Figure 2 marinedrugs-21-00289-f002:**
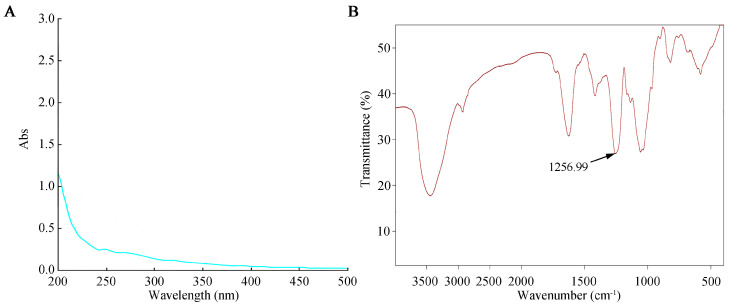
Ultraviolet and infrared spectroscopic analysis of *Sargassum fusiforme* polysaccharide (SFP 2205). (**A**) Ultraviolet spectrum of SFP 2205. (**B**) Infrared spectrum of SFP 2205.

**Figure 3 marinedrugs-21-00289-f003:**
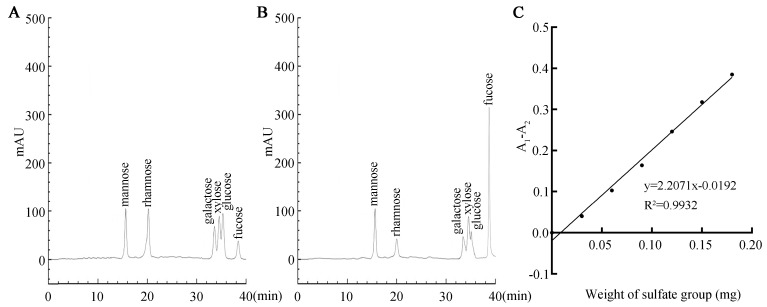
Monosaccharide composition and sulfate group content of *Sargassum fusiforme* polysaccharide (SFP 2205). (**A**) High-performance liquid chromatography (HPLC) of monosaccharide standards. (**B**) Monosaccharide composition of SFP 2205 determined via HPLC. (**C**) Standard curve of the sulfate group.

**Figure 4 marinedrugs-21-00289-f004:**
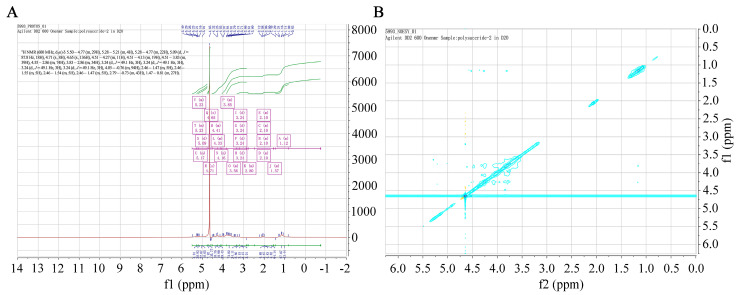
One- and two-dimensional nuclear magnetic resonance spectra of *Sargassum fusiforme* polysaccharide (SFP 2205). (**A**) ^1^H Nuclear magnetic resonance spectrum of SFP 2205. (**B**) Nuclear Overhauser effect spectroscopy spectrum of SFP 2205.

**Figure 5 marinedrugs-21-00289-f005:**
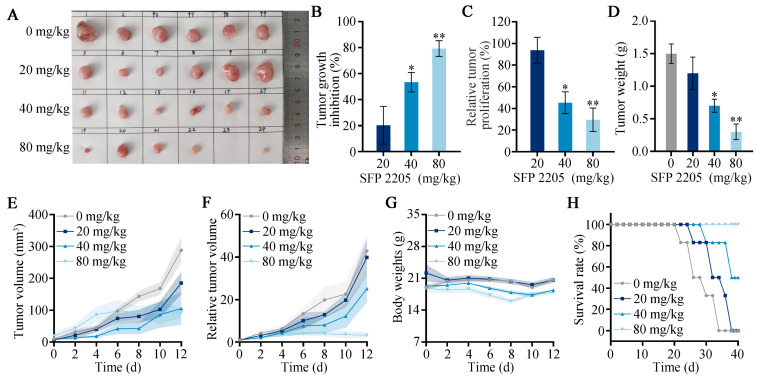
In vivo *Sargassum fusiforme* polysaccharide (SFP 2205) therapeutic effects. (**A**) Representative images of HEL xenograft tumors from each group at day 13. (**B**) Tumor growth inhibition in each group. (**C**) Relative tumor proliferation rate (T/C) in each group. (**D**) Tumor weight of each group. (**E**) Tumor volumes in each group. (**F**) Relative tumor volume (RTV) in each group. (**G**) Body weight of tumor-bearing mice after various treatments at different time periods. (**H**) Survival of mice after each treatment (*n* = 6). * *p* < 0.05, ** *p* < 0.01 compared with the control group.

**Figure 6 marinedrugs-21-00289-f006:**
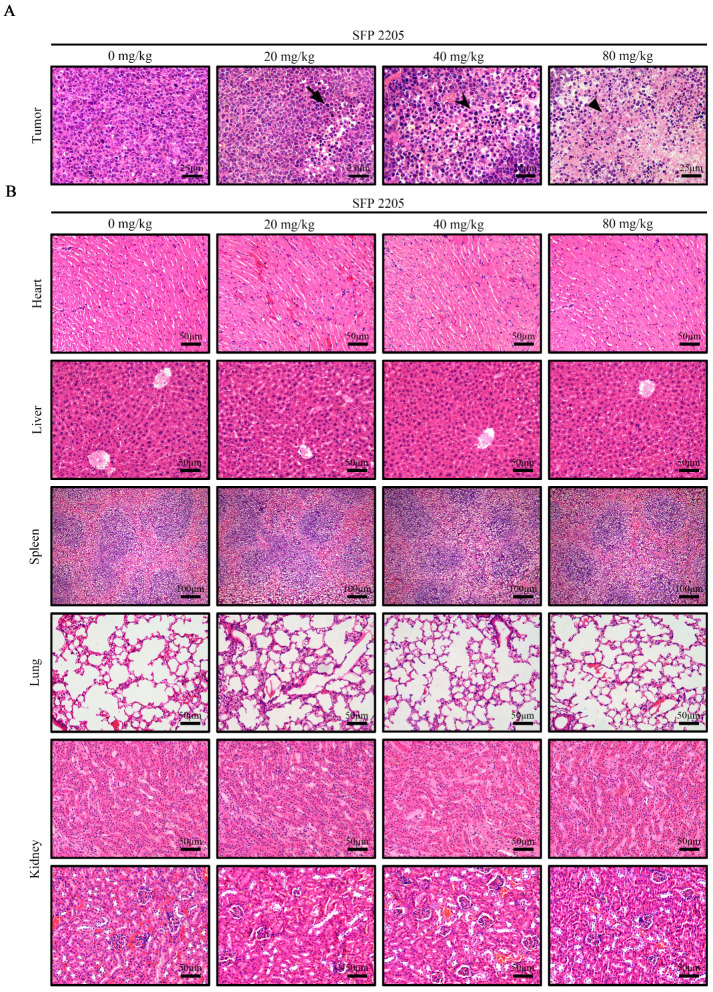
Effects of *Sargassum fusiforme* polysaccharide (SFP 2205) on organ indexes in Balb/c tumor-bearing mice. (**A**) Morphological analysis of tumor tissues via hematoxylin and eosin (H&E) staining. Scale bars: 25 μm. 🠋 indicates nuclear shrinkage, ⮟ indicates an increased cytoplasmic area, and ▼ indicates fatty degeneration. (**B**) H&E staining of major organs from each group. Scale bars: 50 μm.

**Figure 7 marinedrugs-21-00289-f007:**
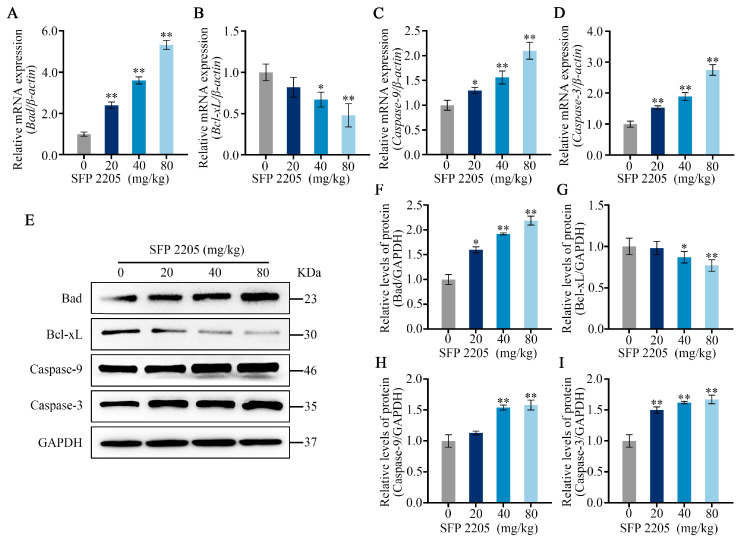
Effects of *Sargassum fusiforme* polysaccharide (SPF 2205) on apoptosis-related gene expression and proteins in tumors of HEL-bearing mice. (**A**–**D**) Apoptosis-associated genes in the tumors were evaluated by quantitative real-time-PCR with *β-actin*, *Bcl-xL*, *Bad*, *Caspase-9*, and *Caspase-3*, respectively. (**E**) Apoptosis-associated protein expressions in tumor tissues. Proteins were analyzed via Western blot with anti-GAPDH, Bcl-xL, Bad, Caspase-9, and Caspase-3 antibodies. (**F**–**I**) Blots were scanned, and Bcl-xL, Bad, Caspase-9, Caspase-3, and GAPDH expressions were measured by densitometric analysis, respectively. The ratios for these proteins are shown. * *p* < 0.05, ** *p* < 0.01 compared with the control group.

**Figure 8 marinedrugs-21-00289-f008:**
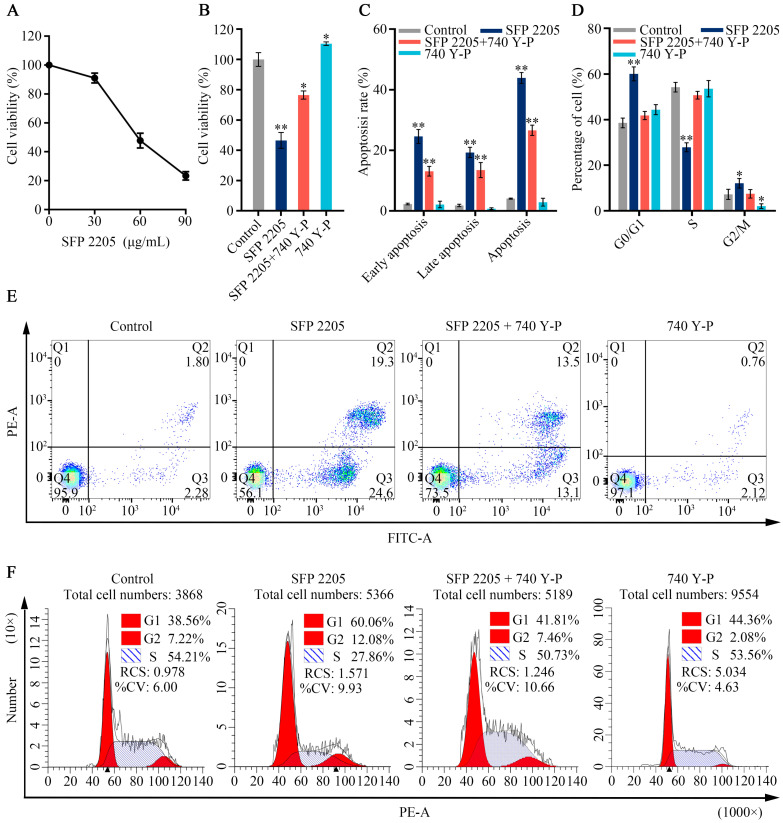
Effects of the PI3K/AKT signaling pathway agonist, 740 Y-P, in HEL cells. (**A**) Inhibitory effect of *Sargassum fusiforme* polysaccharide (SPF 2205) on the growth of HEL cells relative to RPMI 1640-treated control cultures. (**B**) Analysis of *Sargassum fusiforme* polysaccharide (SPF 2205) and 740 Y-P effects on HEL cell apoptosis. (**C**) Bar chart of percentages of HEL cell apoptosis. (**D**) Quantitative analysis of cell cycle distribution of HEL cells. (**E**) Detection of the apoptotic degree of HEL cells treated with SPF 2205 or 740-YP through annexin V- fluorescein isothiocyanate (FITC) and propidium (PI) staining. Q1 indicates dead cells (FITC annexin V-negative and PI-positive); Q2 indicates cells in end-stage apoptosis (FITC annexin V-positive and PI-positive); Q3 indicates cells undergoing apoptosis (FITC annexin V-positive and PI-negative); Q4 indicates viable cells not undergoing apoptosis (FITC annexin V-negative and PI-negative). (**F**) Cell cycle analysis of HEL cells treated with SPF 2205 or 740-YP detected via flow cytometry. * *p* < 0.05, ** *p* < 0.01 compared with the control group.

**Figure 9 marinedrugs-21-00289-f009:**
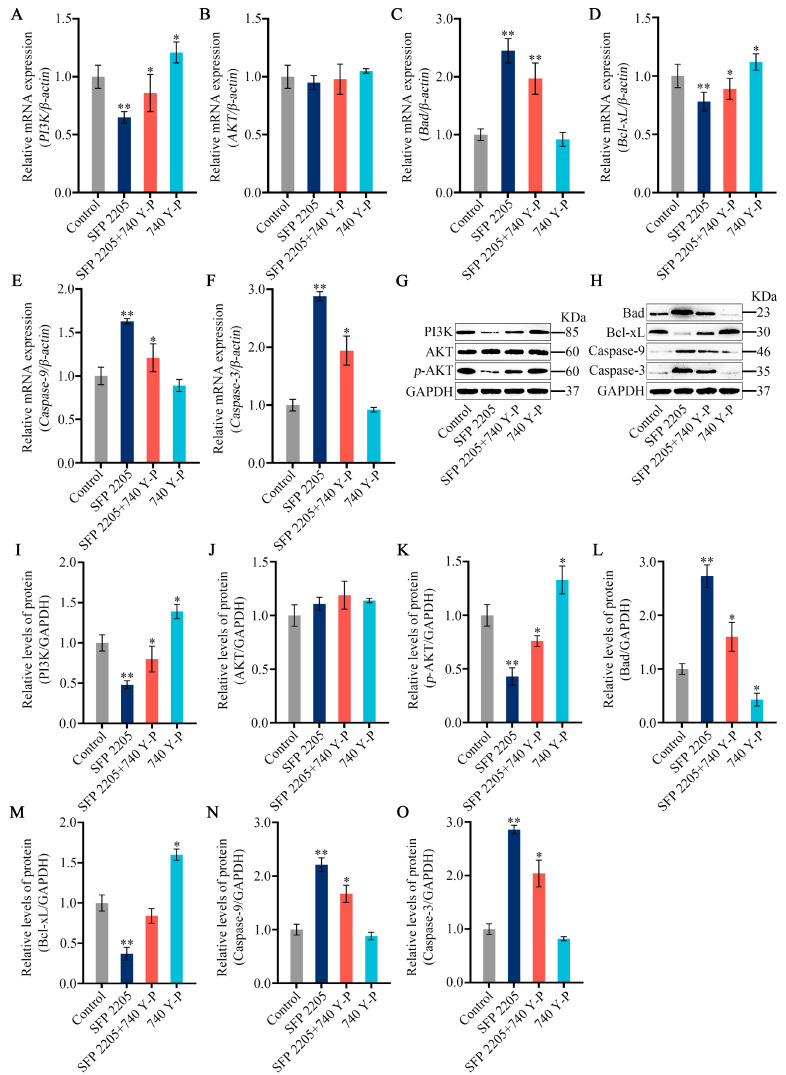
PI3K/AKT signaling pathway mediated the effects of *Sargassum fusiforme* polysaccharide (SPF 2205) on HEL cells. (**A**–**F**) Apoptosis-associated genes in HEL cells were evaluated via quantitative real-time-PCR with *β-actin*, *PI3K*, *AKT*, *Bcl-xL*, *Bad*, *Caspase-9* and *Caspase-3*, respectively. (**G**,**H**) Apoptosis-associated protein expression in HEL cells. Proteins in tumors were evaluated by Western blot with anti-GAPDH, PI3K, AKT, *p*-AKT, Bcl-xL, Bad, Caspase-9, and Caspase-3 antibodies. (**I**–**O**) Blots were scanned, and PI3K, AKT, *p*-AKT, Bcl-xL, Bad, Caspase-9, Caspase-3, and GAPDH expressions were measured by densitometric analysis, respectively. The ratios for these proteins are shown. * *p* < 0.05, ** *p* < 0.01 compared with the control group.

**Table 1 marinedrugs-21-00289-t001:** Primers used in the quantitative real-time polymerase chain reaction assay.

Genes	Sense Primers (5′-3′)	Antisense Primers (3′-5′)
*Caspase-3*	TCCACGAGCAGAGTCAAA	ACACACTTGAACCAACCG
*Caspase-9*	CCTGTATCATCCCCACCCT	CACAAGGTTCCAGAGCCG
*Bcl-xL*	GGAGCTGGTGGTTGACTTTCT	CCGGAAGAGTTCATTCACTAC
*Bad*	CCCAGAGTTTGAGCCGAGTG	ATCCCTTCGTCCTCCGT
*AKT*	CGAGGAGGAGGTGTATCA	ATGGCTTGCACGGAAATGGC
*PI3K*	AACGAGAACGTGTGCCATTTG	AAGCTGTCGTACGGTTAGAGA
*β-actin*	CCTGGCACCCAGCACAAT	GGGCCGGACTCGTCATAC

## Data Availability

The data presented in this study are available on request from the corresponding author.

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
