# Peer review of "Anti-Leukemia Activity of Polysaccharide from Sargassum fusiforme via the PI3K/AKT/BAD Pathway In Vivo and In Vitro"

_marinedrugs, 2023, doi:10.3390/md21050289_

Round 1

Reviewer 1 Report

The manuscript, “Anti-leukemia activity of polysaccharide from Sargassum fusiforme via the PI3K/AKT/BAD pathway in vivo and in vitro”, assessed its medicinal impacts in Balb/c nude mice to verify whether SFP 2205 could inhibit HEL tumor growth in vivo. This manuscript did a lot of work on mice and cell experiments to evaluate the treatment effects of this kind of polysaccharides but presented some problems. Therefore, this paper may need a minor revision. Comments are as follows:

1.    Line 18, it would be more accurate to change the “mole ratio” to “monosaccharides composition”. The formation of mole ratios usually be like “Xyl/Rha/Fuc/Man/Gal =1.00:0.47:27.22:1.28:9.12”

2.    This document(doi.org/10.3390/foods12051115 )is closely related to this manuscript. It is recommended to cite this document.

3.    In lines 246-248, the G0/G1 phase was blocked by SFP 2250 while the 740 Y-P reversed the effect. So, the result of “740 Y-P blocked G0/G1 phase arrest and induced SFP 2205-associated apoptosis” may need to check. And I suggest the author cite some similar papers to support your results and discussion.

4.    What are the total cell numbers, CV%, and RCS in Figure 8E?

5.    Why the author used 740Y-P but did not set a control group?

6.    Line 443, the cells were treated with various concentrations of SFP 2205, but the result only showed one concentration, and the concentration were not clear.

Minor editing of English language required. For example, polysaccharide or polysaccharides, used in MS.

Reviewer 2 Report

The manuscript “Anti-leukemia activity of polysaccharide from Sargassum fusiforme via the PI3K/AKT/BAD pathway in vivo and in vitro” describes the effect of sulfated heteropolysaccharide isolated from brown alga Sargassum fusiforme, SFP 2205, on proliferation and apoptosis induction of leukemia cells and HEL tumor-bearing mice. The authors represented results of the structural characterization of polysaccharide, animal study, genes and proteins expression of mitochondrial apoptosis and PI3K/AKT/mTOR pathways. This is a methodological accurate study and well-written paper containing interesting results which merit publication. For the benefit of the reader, however, a number of minor points need to be corrected. There are given below:

Abstract (Line 14): the word “stimulates” should be exchanged to “stimulated” one.

Keywords: the word “mitochondrial apoptosis” should be added.

Line 79: …determined the sulfation degree or content of sulfate groups, but not sulfated acid concentration.

Figures 8, 9: Is it possible to add graph of 740 Y-P alone to the figures, that readers can judge about the effect of SFP 2205+740 Y-P?

"In vitro" and "in vivo" should be in Italic in the title of the manuscript and  through the text.

Minor editing of English language required.
